# Effects of Structural Reward Shaping on Biophysical Properties in RL-Trained Plasmid Generators

**McClain Thiel** [1]  **Angus G. Cunningham** [1]  **Chris P. Barnes** [1]

## Abstract

We compare the efficacy and distributional effects of supervised fine-tuning (SFT) and reinforcement learning (RL) post-training for PlasmidGPT, a foundation model for whole-plasmid generation, using Group Relative Policy Optimization (GRPO) for the RL model. Using a biologically motivated reward function encoding functional annotations, length constraints, and repeat penalties, the RL model achieves a 71.6% quality-control pass rate across 8 prompts on 4,000 sequences, compared to 4.3% for the pretrained baseline and 11.0% for SFT. A five-model reward ablation identifies the cassette arrangement bonus, which rewards correct promoter→CDS→terminator ordering, as the critical reward component. Rejection-sampling baselines indicate that the gain is not recovered by sampling more heavily from the base model. Beyond directly optimized features, RL-generated sequences converge toward real plasmid distributions in 3-mer composition and minimum free energy density, neither of which is directly optimized by the reward function. Minimum free energy density independently converges to the real-plasmid regime under both SFT and RL despite these being parallel post-training paths. On a small curated hold-out set, RL improves continuation log-likelihood over the pretrained baseline on all 29 held-out sequences (mean $\Delta = +0.83$ nats).

## 1. Introduction

Plasmids are extrachromosomal DNA sequences that are often found in bacteria and are capable of replication independent of a host genome (Lederberg, 1952). These genetic elements are ubiquitous in biotechnology, serving as the primary vectors for protein expression, gene editing, and emerging DNA therapeutics (Prather et al., 2003; Kutzler & Weiner, 2008). Despite their widespread utility, plasmid engineering remains a complex, high-dimensional optimization problem. Traditional workflows are cost-intensive and heuristic-driven, often requiring iterative cycles of manual sequence editing and experimental validation (Oliveira et al., 2009; Meng & Ellis, 2020). Suboptimal architectures with incompatible regulatory elements or unstable repeat regions can lead to metabolic burden, reduced expression efficiency, and manufacturing bottlenecks (Brophy & Voigt, 2014; Wu et al., 2016).

Current approaches to plasmid design rely heavily on tacit domain knowledge and piecemeal assembly of genetic parts. Designers must simultaneously optimize for competing objectives such as copy number, transcriptional output, and host viability while navigating the strict biophysical constraints of DNA folding and context-dependent regulatory interactions (Deng et al., 2025; Fung et al., 2025).

Reinforcement learning post-training has proven effective in natural language processing for improving instruction following and reasoning (Ouyang et al., 2022; Shao et al., 2024b). We investigate whether structural reward shaping can similarly improve generative DNA models, applying Group Relative Policy Optimization (GRPO) to the PlasmidGPT foundation model (Shao, 2024a) for whole-plasmid generation. Our model achieves a 71.6% quality-control pass rate compared to 4.3% for the pretrained baseline across an 8-prompt evaluation on 4,000 sequences. Beyond directly optimized features, generated sequences align with real plasmids in 3-mer composition. Both SFT (next-token loss) and RL (GRPO reward shaping starting from Base) independently converge to plasmid-like 3-mer composition and MFE density. Across 29 non-Addgene held-out plasmids, RL outperforms the pretrained baseline on every plasmid for both continuation log-likelihood and CDS-junction surprisal; within this benchmark, RL's reward shaping does not degrade next-token prediction relative to Base.

This study is scoped to engineered *E. coli* expression vectors using the Addgene repository as our target distribution. Natural and environmental plasmids (e.g., PLSDB) are ex-

[1]Department of Cell and Developmental Biology, University College London, London, UK. Correspondence to: Chris P. Barnes <christopher.barnes@ucl.ac.uk>.

*Proceedings of the 43rd International Conference on Machine Learning*, Seoul, South Korea. PMLR 306, 2026. Copyright 2026 by the author(s).

cluded because our reward function and QC pipeline are designed for engineered constructs with known functional annotations.

Our contributions are: (i) the first application of GRPO to whole-plasmid generation, evaluated across 8 diverse prompts on 4,000 sequences; (ii) a five-model reward ablation identifying the cassette bonus as the critical reward component, whose removal drops pass rate from 66.9% to 19.8% at the ablation evaluation temperature ($T = 0.95$; Appendix B); and (iii) rejection sampling baselines demonstrating that RL produces genuine distributional shift rather than selective filtering (Base 4.3% vs. RL 71.6%; Table 1).

Code, trained model weights, and evaluation data are released at `https://github.com/UCL-CSSB/PlasmidRL`.

## 2. Background

### 2.1. DNA Language Models

Foundation models pretrained on large corpora have shown strong transfer to genomic tasks, including variant prediction (Ji et al., 2021), transcription factor binding (Nguyen et al., 2023), and whole-genome generation (Nguyen et al., 2024).

Plasmid DNA has received relatively little attention compared to other sequence types despite its importance in biomanufacturing and research. OriGen (Irvine et al., 2025) introduces a generative model to produce previously undiscovered origins of replication (ORIs) but does not model whole sequences. PlasmidGPT (Shao, 2024a; Cunningham et al., 2025) uses modern language modeling techniques to develop a generative model for whole plasmid sequences, and later work expands on this by synthesizing whole plasmids generated using a fine-tuned version of the PlasmidGPT model (Cunningham et al., 2025).

Recent work has begun applying reinforcement learning to DNA sequence design. TACO (Yang et al., 2025) uses RL to fine-tune HyenaDNA for promoter and enhancer design (short regulatory elements, <1 kb), while GENERator (Wu et al., 2025) is a long-context genomic foundation model whose design applications include cis-regulatory element generation. Neither targets whole multi-gene plasmid constructs of several kilobases. Our work extends RL post-training to whole-plasmid generation, where the model must coordinate multiple functional components simultaneously across several kilobases of sequence.

### 2.2. Plasmid Design

Lab-designed plasmids are short circular DNA molecules (typically 2-15 kb) that must contain multiple functional components arranged in precise configurations. At mini-

mum, a viable plasmid requires: (i) an origin of replication to enable autonomous replication, (ii) a selection marker (e.g., antibiotic resistance gene) for identifying successfully transformed cells, and (iii) a cloning site where genes of interest can be inserted.

The search space of valid plasmids is massive due to combinatorial explosion across components (Naseri & Koffas, 2020). Each component class can contain hundreds to thousands of variants, and additional regulatory elements (promoters, terminators, enhancers) and reporters may be required depending on the application. Multiple instances of some components may be necessary, and their ordering and spacing significantly affect function. Beyond sheer combinatorics, designers must navigate complex biological constraints such as compatibility requirements (specific ORIs only function in certain hosts), physical stability issues (repeat regions can fold and bind to each other), and other design challenges (Meng & Ellis, 2020; Brophy & Voigt, 2014).

## 3. Methods

We compare GRPO against two baselines: the pretrained PlasmidGPT model and a supervised fine-tuned (SFT) baseline from prior lab work (Cunningham et al., 2025). That work introduced the curated SFT corpus and bioinformatics QC pipeline, including limited wet-lab validation of sequences passing the screen; we reuse the same QC criteria here and train an RL model optimized for sequence-level structural rewards (Figure 1).

### 3.1. Supervised Fine-Tuning

The SFT baseline follows prior work from our lab (Cunningham et al., 2025). Briefly, supervised fine-tuning was performed on a curated corpus of *E. coli* plasmid sequences assembled from PlasmidScope and Addgene (Li et al., 2025; Addgene, 2024). After deduplication and quality filtering, approximately 15k circular plasmids ($\leq$30 kb) were retained, excluding linear entries, fragments, and incomplete records. Sequences were tokenized using the original PlasmidGPT byte-pair encoding (BPE) DNA tokenizer, where tokens average 20–60 nucleotides; consequently, the 256-token generation limit used during RL corresponds to approximately 5–15 kb of sequence. The pretrained PlasmidGPT model was fine-tuned using an autoregressive next-token prediction objective with gradient accumulation and learning-rate warmup over three epochs.

### 3.2. Reinforcement Learning with GRPO

We implement a configurable reinforcement learning pipeline for plasmid design that uses Group Relative Policy Optimization (GRPO) (Shao et al., 2024b) with a domain-

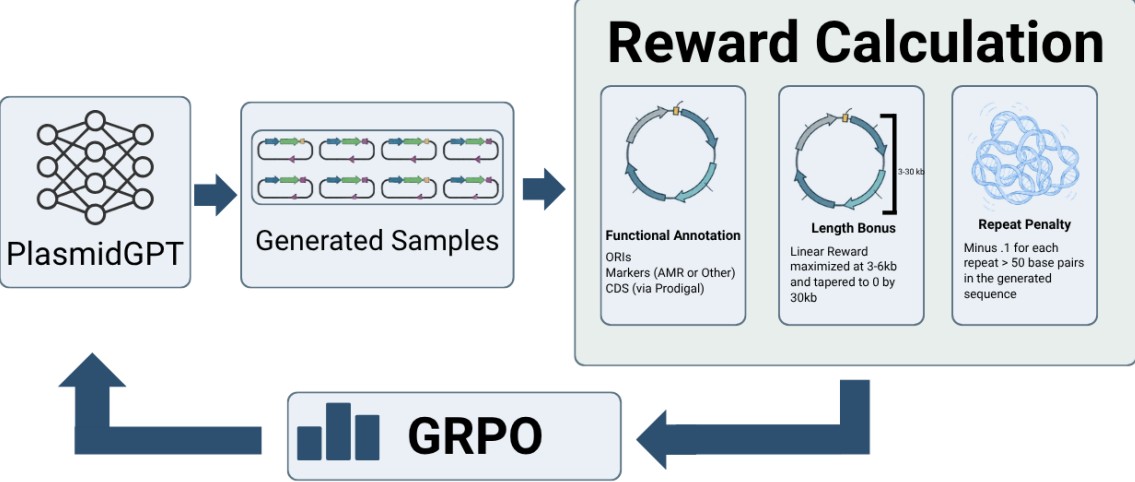

*Figure 1.* Plasmid-RL training pipeline. GRPO is applied to PlasmidGPT using a reward function encoding functional annotations, length constraints, and repeat content.

specific reward function described below. At each training iteration, the model generates a batch of candidate plasmids via autoregressive rollouts conditioned on short nucleotide prompts. Prompts are either stochastic (4–25 bp random seeds, excluding "ATG", used to promote rollout diversity) or structured (partial "cassette" seeds encoding canonical marker genes such as antibiotic-resistance or fluorescent reporters). Each candidate sequence is evaluated via our reward function, which captures structural plausibility, cassette organization, repeat content, and other biologically motivated constraints. GRPO is then applied to update the model parameters using these sequence-level rewards, enabling the policy to progressively shift toward generating plasmids with higher predicted validity.

### 3.3. Reward Function Design

The reward function scores each generated plasmid according to its structural plausibility and expected stability. It is composed of three conceptual components:

**Functional annotation scoring:** Lightweight annotations identify ORIs, promoters, terminators, coding sequences (CDS), and selectable markers, which are then scored according to a configuration designed with subject matter expert (SME) input to reflect biologically reasonable quantities (e.g., exactly one origin of replication and at least one selectable marker). CDS regions are identified using Pyrodigal

(Hyatt et al., 2010; Larralde, 2022), a gene prediction tool that detects coding sequences based on statistical patterns rather than homology search. To encourage coherent gene cassettes, this component also includes a location-aware bonus for promoter $\rightarrow$ CDS $\rightarrow$ terminator arrangements that appear in the correct order and within a reasonable proximity window.

**Length prior:** A length prior favors plasmid sizes within typical experimental ranges preferred for plasmid construction. The length factor is 1 plus an extra bonus inside the acceptable range: sequences between 3 kb and 6 kb receive the full extra length bonus, and this extra bonus tapers to zero at the acceptable-range bounds (2 kb and 30 kb). Sequences outside [2 kb, 30 kb] receive length factor 0.

**Repeat penalty:** A repeat penalty down-weights sequences containing long exact repeats that are associated with instability or recombination, subtracting 0.1 reward for each repeat of length 50 bp or greater.

These terms are combined into a single scalar in [0, 1], yielding a fast and interpretable proxy for "plasmid-likeness" and validity during reinforcement learning. Algorithm 1 gives the full scoring pipeline and Algorithm 2 the cassette-bonus subroutine; component weights and count thresholds are listed in Appendix A.2.

# 4. Experiments

## 4.1. Plasmid Quality Control and Uniqueness

We evaluate model variants across 8 diverse prompts spanning four categories: a minimal start codon (ATG), random short seeds (10 bp, 25 bp), structured cassette prompts (GFP expression cassette), and backbone fragments of varying length (100–300 bp from common plasmid vectors). For each prompt, 500 sequences were generated for a total of 4,000 sequences per model. Prompt strings were excluded from the RL prompt distribution; the underlying biological motifs may still occur in the pretraining or SFT corpora, so evaluation reflects generalization at the prompt level rather than the motif level. Notably, ATG is explicitly excluded from the training prompt distribution (see Appendix A.2), so its inclusion here tests out-of-distribution behavior.

### 4.1.1. VALIDITY ASSESSMENT

In silico plasmid validity was assessed using the bioinformatics quality-control pipeline introduced in our prior lab work (Cunningham et al., 2025), which requires exactly one origin of replication (≥99% identity and coverage via BLAST (Altschul et al., 1990) against a curated ORI database), one or two antimicrobial resistance genes called by AMRFinderPlus (Feldgarden et al., 2021) with 100% identity and coverage, and no internal repeats longer than 50 bp. This pipeline has been evaluated with limited wet-lab validation of screen-passing sequences (Cunningham et al., 2025). While we do not perform wet-lab validation in this work, our focus is on establishing that RL post-training can successfully navigate the plasmid design space in silico, laying groundwork for future conditional generation systems where user-specified designs can be experimentally validated.

### 4.1.2. UNIQUENESS ASSESSMENT

To assess whether generated plasmids represent genuinely new designs rather than minor variants of existing constructs, we compute similarity to known sequences on a subset of generated plasmids using the NCBI BLASTn API (Altschul et al., 1990). Each generated plasmid is assigned to one of three categories based on identity and query-coverage thresholds: sequences with ≥99% identity and ≥95% coverage are classified as **Exists**; those with ≥95% identity and ≥80% coverage are **Similar**; and all others are classified as **Novel**.

This categorization follows large-scale plasmid curation efforts such as PLSDB, which use similar thresholds to deduplicate plasmids before adding sequences to a dataset (Galata et al., 2018).

*Table 1.* Quality metrics across model variants, all evaluated on 4,000 sequences across the same 8 prompts using the QC pipeline of Cunningham et al. (Cunningham et al., 2025) (see §4.1.1). Diversity is the mean pairwise 21-mer Jaccard distance, computed as a pooled distance over all 4,000 sequences (including cross-prompt pairs); per-prompt within-prompt values are reported in Table 9 and are mechanically lower because they exclude cross-prompt pairs. The Addgene row reports the reference panel; QC is not applicable for real plasmids.

| MODEL | QC PASS RATE | DIVERSITY |
|---|---|---|
| ADDGENE (E. COLI) | — | 0.783 |
| BASE | 4.3% | 0.960 |
| SFT | 11.0% | 0.921 |
| RL | 71.6% | 0.596 |

### 4.1.3. DIVERSITY ASSESSMENT

To monitor potential model collapse, we measure the diversity of many samples from the same prompt. Standard text-diversity metrics (e.g., Self-BLEU) are not well-calibrated for biological interpretation on DNA sequences, so we use the mean pairwise Jaccard distance of the 21-mers of each sequence. Diversity of a group of rollouts is calculated as follows:

$$D = 1 - \frac{1}{\binom{n}{2}} \sum_{i=1}^{n} \sum_{j=i+1}^{n} J(S_i, S_j)$$

where $J(S_i, S_j)$ is the Jaccard similarity between MinHash sketches (Broder, 1997) of sequences $i$ and $j$, and $n$ is the number of sequences in the group.

The diversity metric (pairwise Jaccard distance) serves primarily as a model collapse detector rather than a biological validity measure; $D \approx 1$ indicates near-maximal pairwise dissimilarity, as in random or otherwise unconstrained sequence generation, while $D \approx 0$ indicates collapse onto near-identical outputs. We report numerical values in §4.1.4.

### 4.1.4. RESULTS

Reinforcement learning substantially increases the probability of generating plasmids that pass our bioinformatics quality control (QC) pipeline, while supervised fine-tuning provides a more modest improvement. Table 1 summarizes headline QC pass rate and diversity across models.

Aggregated across 8 prompts, the overall QC pass rate rises from 4.3% with the base model to 11.0% with SFT and 71.6% with RL. These are parallel post-training baselines: SFT provides a ∼2.6× improvement over Base, while RL provides a ∼17× improvement over Base under the same 8-prompt evaluation. Per-prompt pass rates range from 14.4% (pUC19 ORI prefix) to 96.6% (CMV enhancer); full per-prompt results are in Appendix D.

*Table 2.* BLASTn novelty assessment (Base $n=22$, SFT $n=28$, RL $n=30$; subset size limited by API throughput). **Exists**: $\geq 99\%$ identity, $\geq 95\%$ coverage. **Similar**: $\geq 95\%$ identity, $\geq 80\%$ coverage. **Novel**: otherwise.

| Model | Exists | Similar | Novel |
|---|---|---|---|
| Base | 0.0% | 9.1% | 90.9% |
| SFT | 0.0% | 3.6% | 96.4% |
| RL | 0.0% | 33.3% | 66.7% |

*Table 3.* Rejection sampling success rate at varying $K$: fraction of $M=50$ trials in which $\geq 1$ of $K$ samples passes QC. Each model is sampled at its sweep-optimal temperature for this protocol (Base $T=1.0$, SFT $T=1.0$, RL $T=1.15$). The $K=1$ values are drawn from a separately resampled rejection pool and are not identical to the 8-prompt headline pass rates in Table 1: Base $K=1$ (4.3%) coincidentally matches its headline, SFT $K=1$ (9.8%) differs from the headline 11.0%, and RL $K=1$ (76.8%) differs because the rejection protocol samples RL at $T=1.15$ while the headline uses $T=1.0$.

| | $K=1$ | $K=4$ | $K=16$ | $K=64$ |
|---|---|---|---|---|
| Base | 4.3% | 14.5% | 38.8% | 54.5% |
| SFT | 9.8% | 36.3% | 76.3% | 99.3% |
| RL | 76.8% | 95.0% | 99.0% | 100.0% |

The diversity metric (pairwise Jaccard distance) confirms that RL concentrates probability mass on higher-quality regions without collapsing to identical outputs: RL diversity is 0.596 compared to 0.960 (near-maximal) for the base model, where the high diversity reflects unconstrained, mostly invalid generation rather than functional variety. For reference, the same metric computed over 1,087 *E. coli* expression vectors from Addgene yields 0.783 (reported in Table 1), and 0.852 over a random 500-plasmid sample from the full Addgene repository (not restricted to *E. coli* vectors). RL diversity (0.596) sits below the designed-plasmid reference but well above trivial collapse, consistent with the model concentrating probability mass on the valid subspace of plasmid architectures rather than reproducing arbitrary sequence variation.

Applying the BLASTn novelty assessment from Section 4.1.2 to a random subset of generated plasmids (Table 2), RL produces no exact matches to known plasmids in the NCBI database; two-thirds are classified as Novel and one-third as Similar. Base and SFT samples are predominantly Novel, but their low QC pass rates (4.3% and 11.0% in Table 1) indicate that this reflects unconstrained, low-validity generation rather than designed novelty. The shift from ~3–9% Similar in Base/SFT to 33% Similar in RL is consistent with RL pulling its outputs toward *E. coli* expression-vector backbones present in the corpus, without copying any single one verbatim.

### 4.2. Reward Ablation Study

To identify which reward components drive the observed improvements, we train five ablation models by selectively disabling individual reward terms. The full results are deferred to Appendix B; we summarize the key finding here: removing the cassette arrangement bonus is by far the most damaging single change, dropping pass rate from 66.9% to 19.8% (at ablation evaluation temperature $T = 0.95$; see Appendix B). The remaining reward terms make smaller contributions to pass rate.

### 4.3. Rejection Sampling Baselines

To confirm that RL produces genuine distributional shift rather than simply filtering low-quality sequences, we compare against rejection sampling at varying $K$ (Table 3). Each cell is the fraction of $M=50$ trials in which at least one of $K$ samples passes QC, drawn without replacement from a 1,250-sequence pool per (model, prompt), giving $M \times 8 = 400$ total trials per (model, $K$).

Across practical low-to-moderate sampling budgets, the success-rate gap between RL and Base/SFT is large. RL at $K=1$ (76.8%) roughly matches SFT at $K=16$ (76.3%), and RL reaches near-saturation by $K=16$ (99.0%). Base never closes the gap on this protocol: even at $K=64$ Base reaches only 54.5%. The $K=1$ row recapitulates the model ordering of Table 1; the $K=\{4, 16, 64\}$ rows quantify how much rejection sampling can recover in each model. SFT's near-perfect saturation at $K=64$ shows that QC filtering can recover rare valid plasmids even when most candidates fail. RL's saturation at $K=16$ shows that reward-shaped post-training makes valid plasmids much more common under direct sampling.

### 4.4. Distribution Comparison

We compute biophysical summary statistics from generated and reference sequences to assess distributional alignment beyond the QC pipeline's binary pass/fail. The reference panel consists of 500 engineered plasmids from Addgene used for protein expression, genome editing, and other applications (see Appendix A.6 for the selection procedure). Statistics include sequence length, GC content, Jensen-Shannon divergence (Lin, 1991) of the 3-mer composition, and minimum free energy density (ViennaRNA (Lorenz et al., 2011)).

Figure 2 shows that RL-generated sequences closely match real plasmids across the displayed metrics, even those not explicitly rewarded. Quantitative values are in Table 6. GC

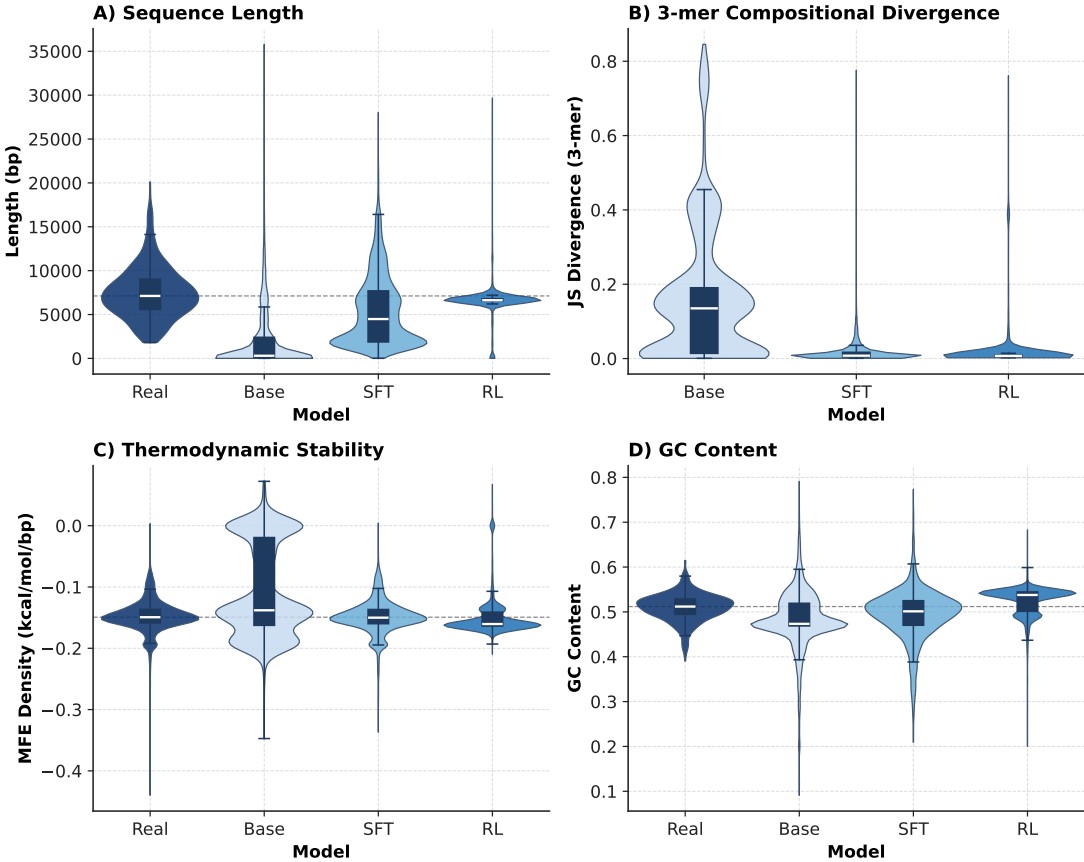

*Figure 2.* Distributional alignment across directly optimized, reward-correlated, and indirectly shaped biophysical metrics.

content and 3-mer composition both shift toward the real-plasmid distribution, with SFT and RL reaching comparable 3-mer divergence despite independent training objectives. RL mean sequence length (6,517 bp) is closest to the Addgene reference (7,470 bp), while Base skews short due to its unconstrained generation.

We categorize these distributional effects into three tiers based on their relationship to the reward function:

**Directly optimized:** Sequence length is explicitly encoded by the reward function. RL mean length is closest to the Addgene reference panel, while Base skews far short and SFT falls between the two (Table 6).

**Reward-correlated:** GC content is not directly optimized, but functional elements selected by the reward (ORIs, coding sequences, resistance markers) carry GC content characteristic of *E. coli* plasmids. RL GC (0.524) slightly overshoots the real-plasmid value (0.510), suggesting that selection pressure on GC-rich functional elements pulls composition above the natural distribution rather than tracking it precisely.

**Indirectly shaped:** 3-mer composition and minimum free

energy (MFE) density are not named by the reward function, but both shift sharply in SFT and RL relative to Base. JSD against the reference 3-mer distribution drops in both fine-tuning paths to near the real-plasmid value (Table 6). MFE density behaves similarly: SFT ($-0.149$) and RL ($-0.149$) both land within noise of the real-plasmid reference ($-0.151$), while Base sits far above ($-0.105$). This convergence from two independent post-training objectives suggests that plasmid-domain training recovers local sequence statistics and a narrow thermodynamic regime without directly optimizing for either, and that the models do not achieve high QC scores by producing physically unusual sequences.

## 4.5. Held-Out Continuation

To evaluate how RL training affects nucleotide-level predictive performance, we measure teacher-forced log-likelihood on real plasmid continuations. For each sequence, we condition on the first 400 nucleotides and compute the per-plasmid mean log-probability of the true next bases under each model using a sliding-window protocol (prefix=400 bp, stride=300 bp, window=100 bp). Table 4 reports the average per-plasmid log-probability for Base and RL on this

*Table 4.* Average per-plasmid log-probability on 29 held-out non-Addgene plasmids (NCBI shuttle vectors, SEVA backbones, and vectors from *Corynebacterium*, *Mycobacterium*, and *Lactobacillus*). RL improves over Base on every plasmid (29/29).

| MODEL | MEAN LOGPROB | ERROR (SEM) |
|-------|--------------|-------------|
| BASE | $-8.99$ | 0.23 |
| RL | $-8.16$ | 0.20 |

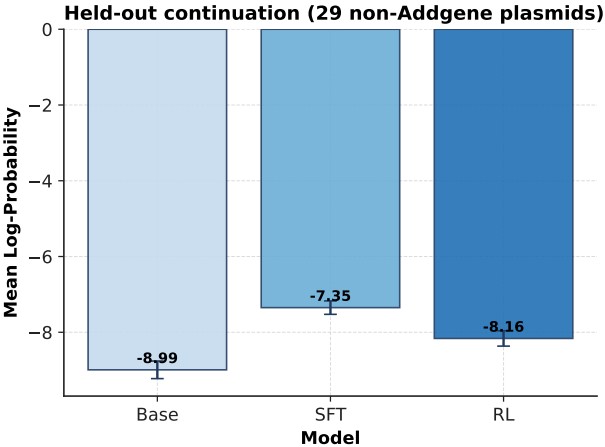

*Figure 3.* Per-plasmid log-probability on 29 held-out non-Addgene plasmids. RL outperforms Base on every sequence (29/29), with consistent improvement across diverse plasmid origins. SFT shown for context.

benchmark.

The RL model shows a robust improvement in mean log-probability compared to the base model (mean $\Delta = +0.83$ nats; paired $t$-test $t = +7.61$, $p = 2.75 \times 10^{-8}$; Wilcoxon signed-rank $p = 3.73 \times 10^{-9}$; Cohen's $d = +1.41$, large effect; RL beats Base on 29/29 plasmids). The corresponding standard errors are 0.23 (Base) and 0.20 (RL). The same pattern holds for CDS-junction surprisal (mean $\Delta = +2.40$ nats, $p = 2.98 \times 10^{-14}$, $d = +2.62$, 29/29). Within this scope we observe no alignment tax relative to Base. Figure 3 shows that this improvement is consistent across all 29 held-out plasmids. We note for completeness that, evaluated on this same held-out set, SFT achieves better continuation log-probability ($-7.35$) than RL. SFT is trained directly on the Addgene corpus and provides a strong predictive baseline that RL does not exceed, but RL strictly improves over Base on this set ($\Delta = +0.83$ nats, 95% CI $[+0.61, +1.05]$, across all 29 plasmids).

### 4.6. CDS-Junction Log-Probability Analysis

For each held-out real plasmid, we identify annotated promoter-to-CDS junctions from pLannotate calls (McGuffie & Barrick, 2021) and compute the model's mean

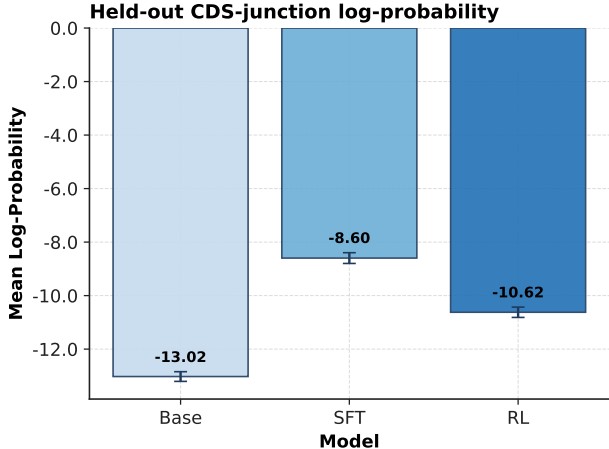

*Figure 4.* CDS-junction log-probability on held-out plasmids. RL achieves higher log-probability (lower surprisal) than Base despite being trained on structural rewards only.

per-token log-probability over a 200 bp window centered on the downstream CDS start. We refer to this evaluation as CDS-junction log-probability (equivalently, negative surprisal). The training reward uses Pyrodigal, the Cython-accelerated Python interface to Prodigal (Hyatt et al., 2010; Larralde, 2022), to detect CDS boundaries on *generated* sequences and reward correctly placed coding regions; the CDS-junction evaluation is computed on *real* plasmids from pLannotate-derived promoter/CDS junctions and is not used by the reward.

The RL-trained model achieves higher CDS-junction log-probability (lower surprisal) on held-out real plasmids than the pretrained model (Figure 4), suggesting that the structural reward sharpens the model's distribution around functional coding boundaries rather than collapsing it onto reward-only patterns.

## 5. Discussion

### 5.1. Effects of Structural Reward Shaping on Biophysical Properties

The central finding of this work is that a structurally motivated reward function encoding functional annotations, length constraints, and repeat penalties produces biophysical alignment with real plasmids across properties at varying degrees of separation from the reward signal. The ablation study (Table 7) demonstrates that the cassette bonus, which rewards correct promoter→CDS→terminator ordering, is the critical component, and that individual reward terms are insufficient in isolation.

The three-tier taxonomy introduced in §4.4 shows that the biophysical effects of RL post-training range from directly

optimized properties (length, component presence) through reward-correlated byproducts (GC content) to indirectly shaped properties (3-mer composition) that arise from the interaction of structural constraints rather than any single reward term. Using the ablation values in Table 7, MFE forms an approximately monotonic gradient that tracks reward richness: the full reward yields the most stable sequences ($-0.149$, matching the Addgene reference $-0.151$ in Table 6), removing individual reward terms produces intermediate MFE values ($-0.141$ to $-0.131$), and the single-component variants (length-only $-0.126$, CDS-only $-0.103$) sit closest to Base ($-0.105$). This suggests that real-plasmid-like thermodynamic stability accumulates as multiple structural constraints compose; no single reward term reproduces it in isolation.

Held-out continuation log-likelihood does not degrade relative to the pretrained baseline on this benchmark (§4.5): RL outperforms Base on 29/29 non-Addgene plasmids for continuation and 29/29 for CDS-junction surprisal, with reduced cross-plasmid variance (std $1.25 \rightarrow 1.09$). RL does underperform supervised fine-tuning on this same set, so we do not claim that RL improves predictive performance over SFT or across all plasmid domains. Within the engineered shuttle-vector scope of the held-out set (*Corynebacterium*, *Mycobacterium*, *Lactobacillus*, and SEVA backbones), RL's structural reward shaping strictly improves next-token prediction over Base on all 29 plasmids (mean $\Delta = +0.83$ nats, 95% CI $[+0.61, +1.05]$, Cohen's $d = +1.41$).

RL post-training has improved reasoning, instruction following, and preference alignment in natural language processing, motivating its application to biological sequence generation. The analogy has important limits. In the common NLP post-training pipeline, RL often follows supervised fine-tuning; here, RL starts directly from the pretrained model and is evaluated as an alternative post-training path alongside SFT. In NLP, reward signals typically encode human judgments about text quality; here, rewards encode structural constraints grounded in molecular biology. Our results suggest that this grounding is advantageous: the reward function selects for properties it does not explicitly encode (3-mer composition, thermodynamic stability), consistent with the hypothesis that structural constraints in DNA are more tightly coupled to biophysical outcomes than stylistic preferences are to text quality. However, biological sequence design ultimately requires function at the organismal level: protein expression, host viability, and metabolic compatibility are all properties that cannot be assessed from sequence structure alone. This study evaluates only in silico structural properties. Bridging the gap between structural validity and biological function in a host organism remains the central open problem for RL-guided biological design.

## 5.2. Implications

Structural reward shaping is a viable approach for constrained biological sequence design. The five-model ablation and rejection sampling baselines demonstrate that RL produces genuine distributional shift: the base model's distribution contains very few valid plasmids regardless of sample count, while RL concentrates probability mass on functional regions of sequence space. This suggests that for generation tasks with verifiable structural constraints, RL post-training offers substantial advantages over supervised fine-tuning at low sampling budgets and over naive sampling strategies. At very large sampling budgets, rejection sampling can recover rare valid SFT outputs, but at much higher sample cost.

## 5.3. Limitations

No generated sequences have been experimentally validated. While our QC pipeline has been validated as a proxy for synthesis success in prior work (Cunningham et al., 2025), wet-lab validation remains the primary limitation and is required before any practical deployment claims can be made.

RL optimization trades diversity for quality. pLannotate (McGuffie & Barrick, 2021) annotation of the 4,000 generated sequences shows two *E. coli* replication origin families with material representation (see Appendix C): the ColE1/pMB1/pBR322/pUC family (detected in 94.6% of sequences) and p15A (12.9%), with roughly a dozen other origin classes appearing in trace quantities ($\leq 0.4\%$ each). Individual sequences may carry multiple detectable origin signatures, so percentages exceed 100%. This pattern reflects the dominance of ColE1-based backbones in the pretraining corpus: RL converges on the pretraining-dominant ColE1 distribution, as expected given that ColE1 is the most probable valid replication origin in the Addgene-like engineered-plasmid distribution that PlasmidGPT was trained on. The reward function amplifies this existing bias rather than introducing it. The distribution narrows but does not collapse. The reduced diversity does limit the model's utility for applications that require alternative copy-number regimes or host ranges; conditional generation that exposes the ORI choice as a control input (Section 5.4) is the natural mitigation.

This study is scoped to engineered *E. coli* expression vectors using Addgene as the target distribution. The reward function and QC pipeline assume known functional annotations, so novel functional elements (e.g., previously uncharacterized ORIs) will not be detected or rewarded. Natural and environmental plasmid diversity (e.g., PLSDB) is not addressed.

### 5.4. Future Work

We plan to develop conditional generation capabilities where users can specify desired plasmid properties (e.g., "express protein X in *E. coli* with high copy number"), which would both increase practical utility and promote sample diversity through structured prompting.

This work tests only GRPO; whether the same structural reward landscape produces comparable results under alternative RL algorithms (e.g., PPO, DPO, REINFORCE-style estimators with off-policy correction) is an open question. A controlled comparison across post-training algorithms on the same reward and base model would help separate algorithm-specific behavior from properties of the reward itself.

## 6. Conclusion

We demonstrate that GRPO post-training with a structurally motivated reward function achieves 71.6% QC pass rate on whole-plasmid generation compared to 4.3% for the pretrained baseline, evaluated across 8 prompts on 4,000 sequences. A five-model ablation identifies the cassette bonus as the critical reward component, and rejection sampling baselines confirm genuine distributional shift. Beyond directly optimized properties, generated sequences align with real *E. coli* expression vectors in 3-mer composition. These effects can be understood through a three-tier taxonomy distinguishing directly optimized, reward-correlated, and indirectly shaped properties of reward-based optimization in biological sequence space. Additionally, minimum free energy density converges to the real-plasmid regime under both SFT and RL independently, despite these being parallel post-training paths with different objectives. Under the MFE metric tested here, structural reward shaping does not appear to achieve high QC scores by producing physically unusual sequence ensembles.

## Acknowledgements

We thank Linda Dekker for discussions and domain input that helped shape the reward function design. McClain Thiel acknowledges support as an Encode Fellow through Pillar VC's Encode Fellowship, part of Pillar VC's Encode: AI for Science initiative. The Encode Fellowship is backed by Pillar VC and powered by ARIA and DSIT.

## Impact Statement

This paper presents work whose goal is to advance the field of computational biology and machine learning for biological sequence design. The ability to generate novel, valid plasmid sequences could accelerate research in synthetic biology, biomanufacturing, and therapeutic development.

While the immediate applications are primarily beneficial for scientific research, we acknowledge the dual-use nature of synthetic biology tools. The methods described here are limited to generating plasmid sequences based on patterns in existing databases and do not enable the design of harmful organisms without substantial additional effort and expertise. We believe the benefits to research efficiency and accessibility outweigh the potential risks, particularly given existing biosafety regulations and oversight mechanisms in synthetic biology research.

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

## A. Training Configuration Details

### A.1. Supervised Fine-Tuning Hyperparameters

- **Batch size:** 1

- **Learning rate:** $5 \times 10^{-5}$ with 500 warmup steps

- **Optimizer:** AdamW

- **Epochs:** 3

- **Gradient accumulation steps:** 8

- **Hardware:** single NVIDIA L4 GPU

### A.2. Reinforcement Learning (GRPO) Hyperparameters

Values below are Optuna-selected and live in the released `plasmidrl/config.py` package module.

- **Per-device training batch size:** 16

- **Generations per prompt (GRPO group size):** 4

- **Policy learning rate:** $1.91 \times 10^{-5}$ (constant schedule)

- **GRPO $\beta$ (KL coefficient):** $8.85 \times 10^{-4}$

- **GRPO $\epsilon$ (clip range):** 0.265

- **Loss type:** BNPO, reward scaling on

- **Max prompt / completion tokens:** 1024 / 256

- **Epochs:** up to 20; training terminated earlier on reward plateau or collapse

- **Training prompt types:** random 4–25 bp seeds (excluding ATG) and structured cassette seeds

- **Hardware:** NVIDIA L40S (Anyscale)

### A.3. Reward Function Configuration

The component weights below were chosen by subject-matter experts on biological grounds before any RL training and were not tuned on the evaluation set. The single Optuna-tuned reward parameter is the length bonus scale (separate from the GRPO hyperparameters in Appendix A.2, also Optuna-selected); the count thresholds and component weights are fixed. Source-of-truth values are in the released `plasmidrl/ablations.py` package module.

- **Origin of replication:** count exactly 1, weight = 1.0

- **Selectable markers:** present/absent binary score, weight = 1.0

- **Promoters:** count 1–5, weight = 1.0

- **Terminators:** count 0–2, weight = 0.5

- **Coding sequences:** count 1–2, weight = 1.0

- **Promoter→CDS→terminator location bonus:** location_bonus_scale = 0.5, proximity threshold 300 bp, top-$K$ = 2 cassettes

- **Length prior:** length factor = 1 plus an extra bonus inside the acceptable range; full extra bonus for sequences in 3–6 kb; extra bonus tapers to zero at the acceptable-range bounds (2 kb and 30 kb); length factor = 0 outside [2 kb, 30 kb]. Bonus scale = 0.7085 (Optuna-selected)

- **Repeat penalty:** $-0.1$ per detected repeat region $\geq$ 50 bp (subject to the $[0, 1]$ reward clip)

- **Count scoring:** components below their minimum count receive proportional partial credit, scaled by 0.5 in punish mode; components above their maximum count receive 0 because violation_penalty_factor = 0. The selectable-marker score is binary: present = 1.0, absent = 0.0.

### A.4. Sampling Parameters

Evaluation temperatures differ between protocols, held constant within each. Top-$p$ is 0.90 and repetition penalty is 1.0 throughout, and generations are capped at 256 BPE tokens ($\approx$ 5–15 kb of DNA). Table 5 summarizes the temperature settings used for each evaluation.

*Table 5.* Sampling temperature per evaluation protocol.

| PROTOCOL | $T$ |
|---|---|
| MAIN 8-PROMPT EVALUATION (TABLES 1, 2) | 1.00 |
| REWARD ABLATION (APPENDIX B) | 0.95 |
| REJECTION SAMPLING TOP-$K$ (TABLE 3), BASE | 1.00 |
| REJECTION SAMPLING TOP-$K$ (TABLE 3), SFT | 1.00 |
| REJECTION SAMPLING TOP-$K$ (TABLE 3), RL | 1.15 |

### A.5. Distribution Comparison Metrics

Table 6 presents summary statistics comparing the distributions of generated plasmids to real plasmids across key biophysical metrics. For scalar summaries, closeness to the Real column indicates alignment; for 3-mer JSD, lower values indicate closer match to the reference 3-mer composition.

---

**Algorithm 1** Plasmid Reward Scoring

1: **Input:** DNA sequence $s$
2: **Output:** Reward score $R \in [0, 1]$
3:
4: Annotate features using PlasmidKit (ORIs, promoters, terminators, markers) and Pyrodigal (CDS) $\rightarrow A(s)$
5: Merge overlapping features of same type (threshold = 0.8)
6:
7: **for** each count-scored component $i \in \{$ori, promoter, terminator, cds$\}$ **do**
8:     Count features $n_i$ in $A(s)$
9:     **if** $n_i < \min_i$ **then**
10:         score$_i \leftarrow 0.5 \times (n_i / \min_i)$
11:     **else if** $n_i > \max_i$ **then**
12:         score$_i \leftarrow$ violation_penalty_factor $\{= 0;$ violating count zeroes the component$\}$
13:     **else**
14:         score$_i \leftarrow 1.0$
15:     **end if**
16: **end for**
17: score$_{\text{marker}} \leftarrow 1.0$ if at least one marker is present, else 0.0
18:
19: **Compute weighted base score and length factor:**
20: base $\leftarrow \sum_i w_i \times$ score$_i$
21: length_factor $\leftarrow$ length score for $s$
22: reward $\leftarrow$ clip(base $\times$ length_factor, 0, 1)
23:
24: **Detect repeat regions:**
25: Find all $k$-mers ($k = 50$) appearing $\geq 2$ times (including reverse complements)
26: Merge overlapping occurrences $\rightarrow \mathcal{R}$ regions
27: repeat_penalty $\leftarrow |\mathcal{R}| \times 0.1$
28:
29: **return** $R \leftarrow$ clip(reward $-$ repeat_penalty, 0, 1)

---

*Table 6.* Distribution comparison metrics (Addgene reference, $n$=500; all models on 4,000 samples, 8-prompt protocol; MFE: Mathews 2004 DNA parameters).

| METRIC | REAL | BASE | SFT | RL |
|---|---|---|---|---|
| MEAN LENGTH (BP) | 7,470 | 1,949 | 5,441 | 6,517 |
| GC CONTENT | 0.510 | 0.484 | 0.493 | 0.524 |
| 3-MER JSD | 0.012 | 0.171 | 0.016 | 0.022 |
| MFE DENSITY | $-0.151$ | $-0.105$ | $-0.149$ | $-0.149$ |

### A.6. Reference Plasmids for Distribution Comparison

For the distribution comparison in Section 4, we use a curated set of 500 engineered plasmids drawn from the Addgene repository. These plasmids span common backbones, selection markers, and use-cases for protein expression and

**Algorithm 2** CDS Cassette Bonus Subroutine

1: **Input:** Annotated features $A(s)$, base $\text{score}_{\text{cds}}$
2: **Output:** Updated $\text{score}_{\text{cds}}$
3:
4: **for** each promoter $p$ **do**
5:     Find nearest downstream CDS $c$ on same strand
6:     Find nearest downstream terminator $t$ on same strand
7:     Award points for:
8:         $p \to c$ correct order on same strand: $+5$ pts
9:         $p \to c$ within 300bp: $+5$ pts
10:        $c \to t$ correct order on same strand: $+5$ pts
11:        $c \to t$ within 300bp: $+5$ pts
12: **end for**
13: Take top $K = 2$ cassettes $\to$ total cassette_points
14: cassette_bonus $\leftarrow 0.5 \times (\text{cassette\_points}/40)$
15: **return** $\text{score}_{\text{cds}} \leftarrow \text{clip}(\text{score}_{\text{cds}} + \text{cassette\_bonus}, 0, 1)$

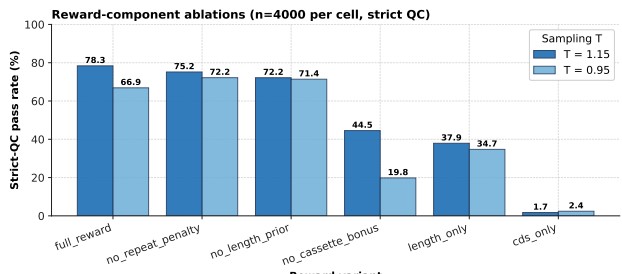

*Figure 5.* QC pass rates across reward ablation variants. Light bars show $T=0.95$ (matching Table 7); dark bars show a $T=1.15$ sensitivity comparison. Removing the cassette bonus produces the largest drop at both temperatures, identifying it as the critical reward component.

*Table 7.* Reward ablation results (all rows at $T=0.95$; $n=4,000$ per model). MFE is mean per-base-pair minimum free energy density (kcal/mol/bp, Mathews 2004 DNA parameters). Diversity is mean pairwise 21-mer Jaccard distance.

| MODEL | PASS | MFE | DIV. |
|---|---|---|---|
| RL (FULL REWARD) | 66.9% | −0.149 | 0.578 |
| NO REPEAT PENALTY | 72.2% | −0.141 | 0.394 |
| NO LENGTH PRIOR | 71.4% | −0.131 | 0.447 |
| NO CASSETTE BONUS | 19.8% | −0.134 | 0.355 |
| LENGTH ONLY | 34.7% | −0.126 | 0.760 |
| CDS ONLY | 2.4% | −0.103 | 0.823 |

genome editing, and serve as a reference panel for evaluating how closely model-generated plasmids match real engineered constructs.

This 500-plasmid distribution panel is distinct from the Addgene diversity references in Table 1: the Addgene (E. coli) row uses 1,087 E. coli expression vectors for the same 21-mer Jaccard diversity metric, and the text also reports a random 500-plasmid sample from the full Addgene repository as a broader, diversity-only comparator. These diversity-only panels are not used for the scalar distribution metrics in Table 6.

All minimum free energy (MFE) calculations use ViennaRNA 2.7.2 with Mathews 2004 DNA energy parameters (Lorenz et al., 2011). Sequences $\leq 3,000$ bp are folded directly; longer sequences use a sliding window approach (500 bp windows, 250 bp stride) with averaged MFE density. Plasmids are treated as circular molecules.

## B. Reward Ablation Study

To identify which reward components drive the observed improvements, we train five ablation models by selectively disabling individual reward terms. Each model is evaluated on 4,000 sequences using the same 8-prompt protocol as the main evaluation. Table 7 presents the results and Figure 5 visualizes the pass-rate column.

The cassette bonus, which rewards correct promoter→CDS→terminator ordering and proximity, is the single most important reward component. Removing it drops pass rate from 66.9% to 19.8% and also collapses diversity to 0.355, the lowest of any ablation. The repeat penalty and length prior contribute smaller amounts to pass rate but reduce diversity substantially when removed (0.394 and 0.447 respectively, vs. 0.578 for the full reward). Training with only the CDS reward (2.4%) or length reward alone (34.7%) yields substantially lower pass rates, confirming that the full reward function's components act synergistically.

The full-reward row uses the same RL model as Table 1 and Table 6, sampled at $T=0.95$ rather than $T=1.0$. Pass rate and diversity differ modestly with sampling temperature (66.9% vs. 71.6%; 0.578 vs. 0.596); MFE is essentially unchanged (−0.149 in both tables).

MFE values across ablations are discussed in §5.1.

Functional diversity (origin-of-replication families) is reported separately in Appendix C from a pLannotate audit over the full 4,000-sequence generations.

## C. Functional Diversity of Generated Sequences

The 21-mer MinHash Jaccard diversity reported in the main text (0.596 for the full RL model) measures sequence-level variation, which on a fixed-backbone plasmid is dominated by cargo and cassette arrangement rather than architectural diversity. A more direct measure of architectural diversity is the distribution over origin-of-replication (ORI) types: the ORI is the strongest determinant of copy number, host range, and compatibility, and is what a downstream user is typically choosing between.

To audit how much functional diversity survived RL training, we annotated all 4,000 GRPO-at-$T{=}1.0$ generations with pLannotate (McGuffie & Barrick, 2021), separately from the validity QC pipeline used in Section 4.1.4. This audit covers all generated sequences, not only those that pass strict QC; a finer-grained audit restricted to the QC-passing subset is left for future work. Table 8 reports the resulting ORI-family counts.

*Table 8.* pLannotate ORI breakdown on 4,000 generated sequences. Counts are unique sequences per origin family (multi-ORI sequences counted once per matched family).

| Origin family | Unique Seq. |
|---|---|
| ColE1 / pMB1 / pBR322 / pUC | 3,782 |
| p15A | 516 |
| IncP | 16 |
| Yeast $2\mu$ | 15 |
| RSF1030 | 12 |
| EBV oriP (variants) | 15 |
| pVS1 (*Pseudomonas*) | 5 |
| S. cerevisiae ARS1 | 5 |
| M13 / pSC101 / SV40 | $\leq$4 each |
| F plasmid, others | $\leq$2 each |

3,797 of the 4,000 sequences carry at least one detectable origin signature. The two well-represented *E. coli* replication origins are ColE1 (in 94.6% of sequences; 3,782 unique) and p15A (in 12.9%; 516 unique); a dozen further origin classes are detected occasionally in trace quantities. We do not count the f1 bacteriophage origin (2,859 sequences) toward replication-strategy diversity because it co-occurs with ColE1 in pUC-style backbones for ssDNA production rather than serving as an independent replication origin.

This pattern reflects the dominance of ColE1-based backbones in the pretraining corpus (predominantly ColE1-based *E. coli* expression vectors, with p15A as the second most common backbone family); RL converges on this distribution because it is already the most probable regime in the pretrained model. RL narrows but does not collapse the distribution over origins. Reduced architectural diversity remains a real limitation for applications that need alternative copy-number regimes or host ranges; conditional generation that exposes the ORI as a control input (Section 5.4) is the planned mitigation.

## D. Per-Prompt QC Pass Rates

Table 9 breaks down the RL model's QC pass rate and sequence-level diversity across the eight evaluation prompts.

*Table 9.* Per-prompt QC pass rates and diversity for the RL model (500 sequences per prompt, $T{=}1.0$).

| Prompt Type | BP | Pass | Div. |
|---|---|---|---|
| Start codon (ATG) | 3 | 88.4% | 0.13 |
| Random seed | 10 | 85.0% | 0.13 |
| Random seed | 25 | 71.0% | 0.10 |
| pACYC184 (p15A) ORI | 100 | 82.0% | 0.10 |
| pUC19 ORI region | 100 | 14.4% | 0.81 |
| KanR cassette | 300 | 80.2% | 0.71 |
| CMV enhancer | 300 | 96.6% | 0.86 |
| GFP cassette | 917 | 55.0% | 0.08 |

