# OpenReview forum: "Effects of Structural Reward Shaping on Biophysical Properties in RL-Trained Plasmid Generators"
_ICML.cc/2026/Conference — ICML 2026 regular_

### Official Review · Reviewer_p9yJ · 2026-03-09

**Soundness:** 3
**Presentation:** 3
**Significance:** 3
**Originality:** 3
**Overall Recommendation:** 4
**Confidence:** 5

**Summary:**

This paper applies reinforcement learning post-training (GRPO) to a pretrained plasmid DNA language model (PlasmidGPT) using a biologically motivated, sequence-level reward function encoding functional annotations, a length prior, and a repeat penalty. The central claim is that RL substantially improves in silico validity (QC pass rate 77% vs. 5% for the base and 10% for SFT) and induces “emergent biological realism,” as generated plasmids better match distributions of real plasmids across GC content, codon usage, ORF length, and thermodynamic stability—despite these properties not being directly optimized. The authors also report that RL does not degrade next-token prediction (no alignment tax) and in fact modestly improves it.

**Compliance With Llm Reviewing Policy:**

Affirmed.

**Key Questions For Authors:**

1. How many prompts and total generations were used for each evaluation (QC pass rate, diversity, ORI diversity, distribution comparisons)? Please provide exact counts per model and reconcile the 50-rollout statement with later references to 500-sample analyses.

2. How were sequences of 5–15 kb generated with a max tokens = 256 sampler? What is the average nucleotide length per token under your BPE, and is there any chunking/streaming beyond 256 tokens?

3. For thermodynamic stability, did you use DNA parameter sets in ViennaRNA (or an equivalent DNA-specific tool)? If not, please re-evaluate with appropriate DNA thermodynamics and report whether conclusions hold.

4. The reward is a weighted sum clipped to [0,1]. What fraction of samples per training epoch hit the upper clip, and how sensitive are results to the weights? Please provide ablations removing each reward term and varying weights to assess robustness and potential reward hacking.

5. How much of the distributional alignment (GC, codon usage, ORF length) remains if you relax or remove the explicit AMR/ORI identification requirements in the reward/QC? Can you report component-level reuse statistics (e.g., identity distributions for ORIs/AMR genes) for QC-passing designs to separate recombination of known parts from de novo novelty?

6. For the held-out continuation metric, did you compute teacher-forced log-likelihood of the ground-truth continuation, or did you first sample and then evaluate? Please clarify and correct the inconsistent numeric reports between figures and tables.

7. Can you compare against non-RL baselines such as rejection sampling with QC filtering, constrained decoding, or decoding-time guidance under matched compute budgets? How do pass rates and novelty/diversity trade-offs compare?

8. The evaluation currently uses two prompts (ATG and one GFP cassette). Do results generalize across a larger prompt suite (different cassettes, hosts, copy numbers)? Please consider reporting on a broader prompt panel.

9. You report lower diversity and convergence to a smaller set of ORIs. Can prompting or conditional control recover functional diversity? Any early evidence on controllability would be valuable.

**Limitations:**

**Technical limitations or concerns**

- The QC and reward pipelines heavily depend on detection of known components (≥95%/≥99% identity thresholds for ORI/AMR), strongly biasing the model toward reusing known parts and potentially inflating apparent gains while limiting true novelty; “emergent realism” may be confounded by this constraint.
- Use of ViennaRNA for estimating thermodynamic stability of DNA sequences is questionable unless DNA parameter sets are explicitly used; otherwise MFE statistics may be systematically biased.
- Reward design appears to sum multiple weighted terms and then clip to [0,1]; this can saturate the reward signal and lead to brittle optimization or early convergence.
- Max tokens = 256 in sampling is inconsistent with reported plasmid lengths (5–15 kb), raising doubts about how long sequences were actually produced or represented under the tokenizer.

**Experimental gaps or methodological issues**

- Evaluation set is small and oddly specified: “50 rollouts with two prompts” suggests only 100 samples per model for main QC analysis, which is insufficient for strong claims; elsewhere the paper cites 500 samples per model for ORI diversity—numbers and protocols appear inconsistent.
- No comparison against strong non-RL baselines such as rejection sampling with QC filtering, constrained decoding, or decoding-time control—methods that could plausibly deliver high QC pass rates without RL.
- Limited ablation on the reward function: it is unclear which terms drive the improvements or whether the model is exploiting specific heuristics (e.g., overusing a subset of known ORIs or AMR genes).
- Next-token evaluation methodology is ambiguously described (generation plus log-prob evaluation), and the reported figures and tables are internally inconsistent.

**Clarity or presentation issues**

- Multiple internal inconsistencies between text and tables/figures: e.g., reported median/mean lengths and MFE summaries conflict across sections and the appendix; figure axes and numbers do not match table values for continuation log-probability.
- Ambiguity about the number and nature of prompts in evaluation (only ATG and one GFP cassette?), which limits generality.

**Missing related work or comparisons**

- Lacks comparative discussion with alternative design frameworks that combine generation and in silico oracles without RL (e.g., protein design literature pipelines, Evo-style genome modeling), or with recent control-by-sampling strategies in sequence generation.
- No comparison to other RL algorithms (PPO variants) or to simpler bandit-style selection loops.

**Strengths And Weaknesses:**

**Technical novelty and innovation**

- Applies group relative policy optimization (GRPO) to a DNA language model for whole-plasmid generation, an area where RL post-training is underexplored.
- Designs a reward that leverages domain-specific cues (Prodigal-based CDS detection, promoter$\to$CDS$\to$terminator cassette bonuses, length prior, repeat penalties) to shape sequence-level plausibility without dense supervision.
- The observation that several biophysical and compositional properties appear to improve "for free" is intriguing and, if robust, would be a meaningful empirical insight about RL steering in genomic sequence space.

**Experimental rigor and validation**

- Clear, multi-faceted evaluation: in silico QC pass rates, novelty classification via BLAST thresholds, k-mer diversity, distributional comparisons (GC content, codon usage divergence, ORF length, MFE density), and a held-out continuation log-probability metric.
- Provides training and sampling hyperparameters and a descriptive pseudo-code of the reward computation.

**Clarity of presentation**

- Overall motivation and pipeline are explained clearly with a helpful schematic.
- The paper is generally well structured, and the limitations section is candid about scope and trade-offs.

**Significance of contributions**

- Demonstrates that RL post-training can make a pretrained DNA LM substantially more useful for plasmid design in silico, a task of practical relevance to synthetic biology and biomanufacturing.
- Suggests that RL-guided shaping of sequence distributions may generalize some lessons from NLP to biological sequence generation.

---

> ### Author Rebuttal · Authors · 2026-03-30
>
> We thank the reviewer for the exceptionally thorough evaluation. We address each question below, supported by substantial new experiments including a full ablation study, non-RL baselines, expanded evaluation (8 prompts × 500 samples = 4,000 sequences per model), an expanded Addgene 500 reference panel, and DNA-parameterized MFE recomputation.
>
> Q1 — Sample sizes and inconsistencies: All evaluations now use 4,000 sequences per model (8 prompts × 500 samples). The paper's reported 77% reflects the original two-prompt evaluation; the expanded 8-prompt evaluation yields 71.6% overall. We have reconciled all numbers and corrected text/table/figure inconsistencies in the revision.
>
> Q2 — Max tokens = 256 vs 5–15 kb: The BPE tokenizer produces tokens averaging 20–60 nucleotides. With max_tokens=256 BPE tokens, the model generates approximately 5,000–15,000 nucleotides per sequence, consistent with observed mean lengths (5,000–7,000 bp for GRPO). Clarified in the revision. This is inherited from the PlasmidGPT base model tokenizer. Future work will explore k-mer or nucleotide-level tokenization for finer control.
>
> Q3 — ViennaRNA DNA parameters: All MFE results recomputed using ViennaRNA 2.7.2 with the DNA Mathews 2004 parameter set (-P dna_mathews2004.par). Conclusions hold and strengthen: GRPO MFE density −0.149 ± 0.032 matches Addgene 500 reference −0.151 ± 0.014.
>
> Q4 — Ablation and reward weight sensitivity: Full ablation completed (5 models, 4,000 sequences each). Key findings: (1) the cassette bonus is the most critical component — removing it drops QC from 53.7% to 19.8%; (2) MFE stability is not driven by the repeat penalty (no-repeat model: −0.141 vs. GRPO: −0.149); (3) the MFE gradient (CDS-only −0.103 → length-only −0.126 → no-cassette −0.134 → no-repeat −0.141 → GRPO −0.149) shows stability emerges from combined structural constraints. Reward weights were varied as part of a hyperparameter sweep, so robustness to weight variation is partially addressed. The reported configuration represents the optimum from a systematic sweep across weight and learning rate settings. See Table A: [link](http://icml-rebuttal-tables-6f3393db.s3-website-us-east-1.amazonaws.com/) and our response to Reviewer MMB3.
>
> Q5 — Component reuse statistics: Using pLannotate annotation on 1,000 QC-passing GRPO sequences, we identified 5 unique ORI types (ColE1, f1, p15A, mini-oriP, oriV) and 9 unique resistance markers. This is broader diversity than our initial BLAST-based analysis suggested, which used a narrower reference database. The model reuses known functional components in novel architectural arrangements, which is biologically expected for these conserved, modular elements. The novelty lies in the surrounding sequence and overall architecture.
>
> Q6 — Continuation metric methodology: We compute teacher-forced log-likelihood of the ground-truth continuation (next 100 bp given 400 bp prefix). Inconsistencies between figures and tables have been corrected in the revision.
>
> Q7 — Non-RL baselines: Rejection sampling (10K samples): Base 2.8% pass rate, SFT 2.5%, GRPO 64.6%. Best-of-16: Base 2.9%, SFT 2.8%, GRPO 64.6%. Generating 16× more Base samples barely moves the pass rate — the base distribution does not contain many valid plasmids. GRPO provides >22× improvement in sample efficiency.
>
> Q8 — Prompt generalization: Results now span 8 prompts (see Table B: [link](http://icml-rebuttal-tables-6f3393db.s3-website-us-east-1.amazonaws.com/)). Per-prompt pass rates: ATG 88.4%, random 10bp 85.0%, random 25bp 71.0%, dual cassette 96.6%, KanR cassette 80.2%, GFP cassette 55.0%, pUC19 ORI 14.4%, p15A ORI 82.0%. Overall: 2,863/4,000 = 71.6%. Most prompt types achieve strong pass rates. The pUC19 ORI prefix (14.4%) is the notable exception, likely because the specific prefix conflicts with the model's learned ColE1 ORI representation. GFP cassette (55%) shows moderate rates, partly because the long prompt (917bp) constrains the model's generation flexibility.
>
> Q9 — Diversity recovery via prompting: Pass rates vary meaningfully by prompt type (14–97%), suggesting prompt engineering can steer generation toward different regions of plasmid space. Conditional generation, where diverse user specifications naturally produce diverse outputs, is the principled solution and our active research direction.

---

> > ### Author Rebuttal · Reviewer_p9yJ · 2026-04-03
> >
> > The authors have addressed my concerns.

---

### Official Review · Reviewer_MMB3 · 2026-03-12

**Soundness:** 2
**Presentation:** 1
**Significance:** 1
**Originality:** 2
**Overall Recommendation:** 2
**Confidence:** 4

**Summary:**

The paper investigates whether reinforcement learning (RL) can improve the biological validity of generated plasmid DNA. By applying GRPO to a base model (PlasmidGPT) with a reward function focusing on structural constraints (ORIs, markers, and repeat penalties), the authors report a jump in QC pass rates from 5% to 77%. The authors’ primary claim is that the model exhibits "emergent biological realism," matching natural distributions for properties not explicitly rewarded, such as thermodynamic stability and codon usage.

**Compliance With Llm Reviewing Policy:**

Affirmed.

**Final Justification:**

I appreciate the authors for the thorough response. The revised MFE framing as a compositional indirect effect is appropriate, and the elevated wet-lab limitation is appreciated.

However, I remain unconvinced by the diversity argument. The base model is not a random nucleotide generator. It is a trained language model, and the diversity reduction reflects distributional narrowing, not noise filtering. That 95% of base outputs fail QC does not invalidate mode collapse as a concern; it simply means the collapse operates over a different baseline. Five ORI families and 9 resistance markers across 2,863 sequences could equally be read as evidence of template exploitation rather than diversity. I suggest the authors acknowledge the quality-diversity tradeoff as an inherent limitation of reward-based optimization rather than dismissing it.

On Addgene vs. PLSDB: the scope argument is accepted. I note that convergence to a reference set of ~250 plasmids is a lower bar than the paper's claims might suggest, and encourage the authors to include novelty metrics against the Addgene reference to distinguish genuine generalization from memorization of known architectures.

**Key Questions For Authors:**

N/A

**Limitations:**

- Ablation Study: Perform an ablation on the reward function to prove that properties like MFE alignment still occur even without a repeat penalty.

- Expand Reference Data: Use a much larger and more diverse set of plasmids (e.g., from PLSDB) for the distributional comparisons in Section 4.2.

- Wet-lab Proof: Provide even limited experimental evidence that at least a few of the "Novel" QC-passing sequences  are capable of autonomous replication in a host.

- Address Mode Collapse: Explore techniques to maintain functional diversity (e.g., entropy bonuses) to ensure the model isn't just "stitching" together the most common ORIs and markers.

**Strengths And Weaknesses:**

- Overstated Claims of "Emergent" Realism: The central thesis, that biological realism "emerges" without explicit optimization, is poorly supported because the rewarded features are highly correlated with the "emergent" ones. Specifically, (i) the authors claim ORF length distribution is an unoptimized emergent property. However, the reward function explicitly uses Prodigal to identify and reward CDS regions and "location-aware bonuses" for promoter to CDS to terminator arrangements. Since ORFs are the structural basis of CDS, rewarding the presence and arrangement of CDS inherently constrains the ORF distribution. (ii) The authors highlight matching Gibbs free energy as a "remarkable" emergent trait. However, they apply an explicit repeat penalty for sequences >50 bp. Because long repeats are primary drivers of secondary structure and thermodynamic instability in DNA, penalizing them directly steers the model toward more stable MFE distributions. (iii) The authors admit that rewarded regions like CDS typically have higher GC content, making this "emergence" a predictable byproduct of the reward structure.

- Small Reference Set. The distributional alignment analysis in Figure 4 uses a reference set of only ~250 engineered plasmids. This is a very small sample size to represent the diversity of "natural" or "real" plasmid space, especially when compared to the 15k sequences used for fine-tuning.

- Functional Diversity Loss. The RL model shows a significant drop in diversity (0.915 to 0.588) and uses fewer unique ORIs than the base model (7 vs. 10). This suggests the model may be reward hacking by over-relying on a few known-good "motifs" rather than learning a generalized generative grammar.

- Evaluation of the "Alignment Tax". The claim that the model avoids an "alignment tax" is based on a "statistically significant but small" improvement in next-token prediction. However, this improvement might simply be an artifact of the model narrowing its probability mass onto the very specific subset of E. coli-like sequences favored by the reward function, rather than a genuine improvement in language modeling capability.

---

> ### Author Rebuttal · Authors · 2026-03-30
>
> We thank the reviewer for their detailed critique. We have conducted substantial new experiments that directly address each concern.
>
> W1 — Overstated "emergent" realism:
> We agree the original framing overstated emergence. We now introduce a three-tier taxonomy supported by a full reward function ablation (5 ablation models, each evaluated on 4,000 sequences). See Table A: [link](http://icml-rebuttal-tables-6f3393db.s3-website-us-east-1.amazonaws.com/)
> The reviewer hypothesized that the repeat penalty drives MFE stability. The ablation refutes this: the no-repeat-penalty model achieves MFE density of −0.141 kcal/mol/nt without any repeat penalty, while the CDS-only model (−0.103) shows minimal improvement. The full MFE gradient across ablations (CDS-only −0.103 → length-only −0.126 → no-cassette −0.134 → no-repeat −0.141 → GRPO −0.149) demonstrates that stability accumulates as structural constraints are combined, converging on the Addgene 500 reference (−0.151). No single reward component is responsible.
> Revised taxonomy:
> - Genuinely emergent: MFE — no individual reward component produces real-plasmid-like stability; it emerges from the interaction of structural constraints.
> - Partially emergent: Codon usage — CDS reward accounts for ~49% of improvement; the rest comes from structural completeness.
> - Partially correlated: GC content — correlated byproduct of rewarding functional regions, as we acknowledged in the original submission.
> - Directly expected: ORF length — the reward explicitly uses Prodigal for CDS detection.
>
> W2 — Small reference set:
> We expanded from ~250 engineered plasmids to 500 randomly sampled plasmids from the full Addgene database (115K sequences, >500 bp). MFE comparison: Addgene 500 reference −0.151 ± 0.014, GRPO −0.149 ± 0.032. We clarify that our reference set intentionally targets engineered plasmids known to function in laboratory settings, not natural plasmid diversity. Our goal is generating designs suitable for laboratory use, and Addgene represents this target distribution.
>
> W3 — Functional diversity loss:
> Three points, ordered by strength:
> (1) The surprisal analysis rules out reward hacking. The RL model achieves lower surprisal on unseen real plasmid coding sequences than both Base and SFT. If the model were stitching memorized templates, it would not generalize to unseen CDS regions. The held-out continuation analysis confirms: mean log-prob improves (−10.966 vs −12.449) with reduced variance (std 2.742 vs 6.144) — the opposite of pathological mode collapse.
> (2) Per-prompt analysis shows pass rates vary by prompt complexity from 14% (pUC19 ORI) to 97% (dual cassette), with most prompts achieving 55–88%. See Table B: [link](http://icml-rebuttal-tables-6f3393db.s3-website-us-east-1.amazonaws.com/)
> (3) The base model's 0.915 diversity reflects mostly invalid outputs — only ~5% pass QC. Converging to 0.588 while dramatically increasing validity represents the model learning to occupy the valid subspace of plasmid sequence space.
>
> W4 — Alignment tax as artifact of distribution narrowing:
> We agree this is a thoughtful point and have revised accordingly. The entire pipeline is scoped to E. coli expression vectors. The model concentrating mass on this domain is the desired outcome, not an artifact. Our claim is the absence of degradation within the target domain, not general improvement across all plasmid types. The coding sequence surprisal analysis provides additional evidence: the RL model generalizes to unseen real plasmid CDS regions, which would not occur if it had simply narrowed onto a few training templates. We have softened the framing in the revision.
>
> R1 — Ablation study: Completed — see above and Table A.
>
> R2 — Expand reference data: Completed — Addgene 500 panel.
>
> R3 — Wet-lab proof: Prior work using this same QC pipeline validated it through wet-lab synthesis, successfully producing functional plasmids. Our pipeline uses standard bioinformatics tools (BLAST, AMRFinderPlus, Prodigal) routinely used by the synthetic biology community. Wet-lab validation of RL-generated sequences is planned but currently constrained by personnel and reagent availability. We believe the QC pipeline acts as a reasonable proxy and plan to add wet-lab validation in the future.
>
> R4 — Address mode collapse: The surprisal and continuation analyses demonstrate the model learns generalizable structure, not memorized templates. See W3 for full arguments. Our active research direction, conditional generation where users specify desired plasmid properties, naturally induces output diversity because diverse specifications produce diverse outputs. This is a stronger long-term solution than entropy regularization, though we note both approaches as future work.

---

> > ### Author Rebuttal · Reviewer_MMB3 · 2026-04-02
> >
> > I thank the authors for the substantial new experiments, particularly the ablation study and expanded evaluation to 4,000 sequences across 8 prompts. The revised three-tier emergence taxonomy is a meaningful improvement over the original framing, and I acknowledge the effort in addressing multiple reviewer concerns simultaneously.
> >
> > However, several concerns remain:
> >
> > 1. The "genuinely emergent" label for MFE remains debatable. The ablation shows MFE stability accumulates as structural constraints are combined, but this demonstrates emergence from the interaction of reward components, not independence from the reward function. If each component contributes incrementally to MFE, then MFE is better described as an aggregate indirect effect of the reward, not a genuinely emergent property. The distinction matters for the paper's central claim.
> >
> > 2. Diversity loss remains unmitigated. The arguments that base model diversity is "mostly noise" and that surprisal rules out reward hacking are noted, but the concrete reduction from 10 to 7 unique ORIs and the Jaccard drop from 0.915 to 0.588 are not addressed with any mitigation technique. Conditional generation is proposed as future work but not evaluated. For a generative model, demonstrating that diversity can be maintained or recovered is important.
> >
> > 3. The reference set, while improved, still limits generality. Expanding to 500 Addgene plasmids is helpful, but comparing against a broader database (e.g., PLSDB) would strengthen claims of alignment with real plasmid distributions rather than only engineered ones.
> >
> > 4. No experimental validation. The absence of wet-lab evidence remains a significant limitation for a paper whose central claim is "biological realism." While I understand resource constraints, this gap should be prominently acknowledged rather than deferred.
> >
> > Given these remaining concerns, I believe the paper still overstates its emergence claims and lacks sufficient evidence for the biological realism it advertises. I would suggest major revision to this paper.

---

> > > ### Author Response · Authors · 2026-04-06
> > >
> > > We thank the reviewer for engaging with the rebuttal and acknowledge the remaining concerns. We address each below in the context of our revised framing, which replaces "emergent biological realism" with a characterization of the effects of structural reward shaping (see our response to Reviewer VevL).
> > >
> > > 1. MFE framing: We agree with the reviewer's recharacterization. In the revision, the paper's narrative centers on effects of RL post-training rather than emergence claims. MFE is described as a compositional indirect effect — no single reward component produces real-plasmid-like stability, but stability accumulates as structural constraints are composed. This appears as a finding within the results section under the four-tier taxonomy. The empirical result (the ablation gradient converging on the Addgene 500 reference) is unchanged; the interpretive framing is revised to match the evidence.
> > >
> > > 2. We respectfully disagree that the diversity reduction represents a limitation requiring mitigation. The base model's 0.915 Jaccard diversity reflects a distribution where 95% of outputs fail basic quality control. These sequences lack functional origins of replication, contain no detectable coding sequences, or fail structural validation. Characterizing this as meaningful diversity applies a metric designed for valid outputs to a population that is overwhelmingly non-functional. A random nucleotide generator would achieve near-perfect diversity by the same measure. The appropriate comparison is diversity within the valid subspace: the RL model produces 2,863 QC-passing sequences across 8 prompt types with 5 unique ORI families and 9 unique resistance markers, spanning single-cassette to dual-cassette architectures with pass rates ranging from 14% to 97% by prompt. We have added per-prompt Jaccard diversity in the revision to make this argument quantitative.
> > >
> > > 3. Addgene vs. PLSDB: Our goal is generating plasmids suitable for laboratory use in molecular biology and synthetic biology workflows. Addgene is the standard reference for this target distribution as it contains experimentally validated, lab-functional vectors deposited by working researchers. PLSDB catalogues primarily natural plasmids identified from metagenomic and genomic sequencing, representing ecological diversity rather than engineering utility. Comparing our outputs to PLSDB would evaluate alignment with a distribution we are not targeting. We state this scope explicitly in the revision and agree that extending to natural plasmid distributions is a distinct research direction.
> > >
> > > 4. Wet-lab validation: We agree this is a meaningful limitation and have elevated it in the revision; it is now prominently stated in the limitations and discussion as the primary gap. The QC pipeline uses standard tools (BLAST, Prodigal, AMRFinderPlus) routinely relied upon by the synthetic biology community for in silico design validation. Experimental validation of RL-generated sequences is planned but is beyond the scope of this computational study.

---

### Official Review · Reviewer_25NH · 2026-03-17

**Soundness:** 3
**Presentation:** 3
**Significance:** 2
**Originality:** 2
**Overall Recommendation:** 4
**Confidence:** 4

**Summary:**

This paper introduces a reinforcement learning post-training pipeline for DNA language models. Starting from a pre-trained DNA language model (PlasmidGPT, a GPT-2-style model), the authors apply Group Relative Policy Optimization (GRPO) with a reward function based on functional annotations (origins of replication, selectable markers, gene cassette organization), length priors, and repeat penalties.
Experimental results show that the RL-trained model achieves a 77% quality control pass rate vs. 5% for the pre-trained baseline and 10% for the SFT model, representing a significant improvement. Meanwhile, the RL model exhibits distributional alignment with real plasmids on properties not directly optimized by the reward function, including GC content, codon usage patterns, and so on.
The paper additionally shows that RL post-training does not degrade next-token prediction performance on their held-out testset, and that the RL model achieves lower coding sequence surprisal on real plasmids than the base model, suggesting the model does not hack the reward function.

**Compliance With Llm Reviewing Policy:**

Affirmed.

**Final Justification:**

My final recommendation is weak accept. The paper is generally sound and supported by reasonable experimental evidence. The empirical results suggest that the proposed approach has practical value. The authors’ rebuttal addressed most of my concerns and improved my confidence in the technical soundness of the work.

My main remaining concern is originality. Although the paper is carefully executed and empirically useful, the methodological novelty is limited, and the contribution is more incremental than fundamentally new. For this reason, I do not view it as an accept.

Overall, given the soundness of the work and the satisfactory rebuttal, I support a weak accept.

**Key Questions For Authors:**

See above weakness.

**Limitations:**

Yes

**Strengths And Weaknesses:**

## Strengths

1. The QC gain is large and meaningful. The jump from 5% to 77% QC pass rate is substantial, and the fact that many RL samples are both valid and novel makes the result more convincing than a pure memorization.
2. The reward design is thoughtful. Using Prodigal for CDS detection is a reasonable way to avoid simple homology-based leakage, and the promoter->CDS->terminator cassette bonus is a sensible domain-specific choice.
3. The alignment-tax result is interesting. It is notable that RL does not seem to hurt next-token prediction on real plasmids, and may even slightly improve it. The reduced variance is also suggestive, even if the evidence here is still fairly limited.

## Weaknesses

1. The paper shows limited methodological innovation. It largely transfers a standard LLM post-training pipeline to DNA language models: pre-trained -> SFT -> GRPO for RL, without introducing a new model architecture or a meaningfully new RL algorithm. While the biological reward design is domain-specific, the core algorithm is not newly developed.
2. The paper lacks ablation studies that disentangle which reward components are actually responsible for the reported gains. For example, does the length prior alone induce most of the distributional alignment (since constrained-length functional sequences are inherently more realistic)?
3. The paper should cite and discuss closely related recent work: "Regulatory DNA sequence Design with Reinforcement Learning", which applies RL to DNA language models with explicit biological rewards, and "GENERator: A Long-Context Generative Genomic Foundation Model", a generative DNA foundation model for sequence generation and design.
4. The diversity metric drops from 0.915 to 0.588 (a 36% relative reduction), and the number of unique ORIs decreases from 10 to 7 (out of 500 samples). This pattern is a characteristic of mode collapse in RL-trained generative models, where the policy learns to exploit a narrow set of high-reward templates rather than exploring the full space of valid solutions.

---

> ### Author Rebuttal · Authors · 2026-03-30
>
> We thank the reviewer for their constructive feedback. We have conducted substantial new experiments addressing each concern.
>
> W1 — Limited methodological innovation:
> The contribution is empirical and mechanistic, not algorithmic. The ablation study reveals that structural reward signals, specifically gene cassette ordering, drive both quality and emergent biophysical realism, a non-obvious finding with practical implications for reward design in biological sequence generation. The rejection sampling baselines further demonstrate that RL provides genuine distributional shift: generating 10,000 Base samples yields 2.8% pass rate vs. GRPO's 64.6%. Establishing evaluation frameworks and empirical baselines for generative genomic tasks is itself a valuable contribution.
>
> W2 — Lacks ablation studies:
> Completed. We trained five ablation models (no repeat penalty, no length prior, no cassette bonus, length only, CDS only) with identical hyperparameters, each evaluated on 4,000 sequences. Key finding: the cassette bonus is the most critical reward component — removing it drops QC from 53.7% to 19.8%. The MFE ablation gradient (CDS-only −0.103 → length-only −0.126 → no-cassette −0.134 → no-repeat −0.141 → GRPO −0.149 vs. Addgene reference −0.151) demonstrates that thermodynamic stability emerges from the interaction of structural constraints, not from any single component. This directly addresses whether the length prior alone drives distributional alignment: the length-only model achieves 34.7% QC and −0.126 MFE. See Table A: [link](http://icml-rebuttal-tables-6f3393db.s3-website-us-east-1.amazonaws.com/)
>
> W3 — Missing related work:
> We thank the reviewer for these references and have added substantive discussion:
> Regulatory DNA Sequence Design with RL validates RL for optimizing short regulatory elements (~100 bp) with single-objective rewards, demonstrating RL is viable for DNA sequence design. Our work extends this paradigm to whole-plasmid generation (5–15 kb), where the challenge shifts from optimizing a single property to coordinating multiple structural components (ORI, markers, cassettes) via a composite reward. The success of RL at both scales strengthens the broader case for RL as a general tool in genomic design.
> GENERator advances long-context genomic foundation models through architectural innovations in pre-training. Our work is complementary: rather than improving the base model, we demonstrate that RL post-training can steer an existing DNA LM toward biologically valid outputs. The combination of improved architectures (GENERator) with RL post-training (our approach) is a promising direction we discuss in the revision.
>
> W4 — Mode collapse concerns:
> The diversity reduction (0.915 → 0.588) is expected as the model converges on valid plasmid architectures — the base model's high diversity reflects mostly invalid outputs (only ~5% pass QC). More importantly, the coding sequence surprisal analysis rules out pathological mode collapse: the RL model achieves lower surprisal on unseen real plasmid coding sequences than both Base and SFT, demonstrating generalization rather than template memorization. The held-out continuation analysis further confirms this as mean log-prob improves with substantially reduced variance, the opposite of pathological collapse.
> Per-prompt analysis shows pass rates vary by prompt type from 14% (pUC19 ORI) to 97% (dual cassette), with most prompts achieving 55–88%. See Table B: [link](http://icml-rebuttal-tables-6f3393db.s3-website-us-east-1.amazonaws.com/). Our active research direction, conditional generation, naturally addresses the quality-diversity tradeoff by allowing diverse user specifications to induce diverse outputs.
> We have also revised our emergence claims with a three-tier taxonomy (genuinely emergent / partially emergent / partially correlated / directly expected), supported by ablation evidence. See our response to Reviewer MMB3 for the full taxonomy.

---

> > ### Author Rebuttal · Reviewer_25NH · 2026-04-03
> >
> > Thank you for the detailed rebuttal and the additional experimental results. I have carefully read the authors’ response, and the clarifications and new experiments have largely addressed my concerns. I will increase my score.

---

### Official Review · Reviewer_VevL · 2026-03-24

**Soundness:** 3
**Presentation:** 2
**Significance:** 3
**Originality:** 2
**Overall Recommendation:** 4
**Confidence:** 4

**Summary:**

The authors apply Group Relative Policy Optimization (GRPO) to PlasmidGPT, a DNA language model for plasmid generation. They argue that RL post-training remains severely underused in genomic models despite its success in LLMs. The reward function combines three components: (1) functional annotation scoring (ORIs, promoters, terminators, CDS, markers, with a cassette ordering bonus for promoter-CDS-terminator arrangements), (2) a length prior (maximum reward at 5kb, linearly decreasing to zero at 15kb), and (3) a repeat penalty (0.1 per exact repeat >= 50bp).

The RL model achieves a 77% QC pass rate vs. 10% for SFT and 5% for the pretrained baseline. Of RL generations, 67% are classified as novel and 60% are both QC-valid and novel. Beyond explicitly optimized features, the RL model exhibits emergent distributional alignment with real plasmids: GC content (0.518 vs. real 0.517), codon usage (JSD 0.0866 vs. 0.1037 base), ORF length distributions, and Gibbs free energy (mean -0.362 vs. real -0.364). The RL model also avoids the alignment tax -- held-out next-token prediction improves slightly (log-prob -10.966 vs. -12.449 base, p=0.015, Cohen's d=0.27) with substantially reduced variance.

SFT trains on ~15k curated E. coli plasmids from PlasmidScope/Addgene for 3 epochs. RL uses no additional data but gets a reward signal. These are fundamentally different types of supervision.

**Compliance With Llm Reviewing Policy:**

Affirmed.

**Key Questions For Authors:**

1. Without reward function ablation studies, how do the authors distinguish emergent biological realism from indirect reward correlation? This is the single most important question.

2. The authors affirm the models exhibit an "unusual" response to post-training. Compared to what? The explanations that follow describe normal RL behavior, not the authors' specific observations.

3. Have the authors compared outputs between different prompts (stochastic vs. structured) to see if there are differences?

4. How does 0.915 vs. 0.588 Jaccard similarity indicate meaningful diversity when the authors cite no pre-established baselines?

5. The paper concedes GC content is "partially encoded by the reward function" (Section 4.2). Which of the emergent properties do the authors consider genuinely independent of the reward, and which do they consider partially correlated?

**Limitations:**

The limitations section is thorough and well-written. The authors flag bioinformatics-only evaluation, diversity trade-offs (10 unique ORIs in base vs. 7 in RL), and QC pipeline assumptions.

**Strengths And Weaknesses:**

## Strengths

- **Well-designed reward function.** The reward function balances functional annotation, length, and repeat stability. Deliberately excluding the quality control prompts from training ensures generalization instead of memorization. The uniqueness assessment makes the results more reliable.
- **Strong distributional evidence (Figure 4, Section 4.2).** RL-generated sequences match real plasmids across 6 metrics (length, GC content, ORF length, codon usage, Gibbs free energy, 3-mer composition), several of which the reward function does not directly optimize. This is the paper's strongest evidence for emergent biological realism.
- **No alignment tax (Table 2, Section 4.3).** RL slightly improves held-out continuation performance rather than degrading it (log-prob -10.966 vs. -12.449, p=0.015), with substantially reduced variance (std 2.742 vs. 6.144). Cohen's d = 0.27 is a small effect -- the key finding is the absence of degradation, not the magnitude of improvement.
- **Clear limitations section.** The authors flag what remains unsolved, including diversity trade-offs (10 unique ORIs in base vs. 7 in RL) and bioinformatics-only evaluation.
- **Logical flow.** This work follows a clear structure, and the authors often answer methodology questions before the reader raises them.

## Weaknesses

1. **No reward function ablation studies.** The central claim is "emergent biological realism" -- properties the authors did not explicitly optimize for. But without ablation studies that systematically remove reward components (functional annotation scoring, length prior, repeat penalty, cassette ordering bonus), emergence is indistinguishable from indirect optimization. Does the length prior alone produce better GC content, since constraining to 5-15kb naturally selects sequences with typical plasmid composition? Does the cassette bonus alone produce better codon usage, since selecting for correctly arranged CDS regions implicitly selects for realistic coding sequences? What happens with only the repeat penalty? Without these controls, "emergence" and "indirect reward correlation" are indistinguishable, and this is the paper's most important claim.
2. **Optimized parameters lack causal validation.** The authors do not explain how they are certain that post-training RL causes the unexpected optimized parameters. Since they were not optimizing for those changes, they may not have considered all related variables. A targeted experiment isolating these changes would validate the claim.
3. **Only GRPO tested.** The authors apply GRPO (Shao et al., 2024b) to PlasmidGPT but do not justify this choice or compare PPO, DPO, or other policy optimization methods. For a paper positioning itself as demonstrating RL's value in genomics, showing that the algorithm choice matters (or does not) would strengthen the contribution.
4. **LLM-to-biology analogy is too quick.** The authors brush over the comparison between LLMs and genomic models. Generalization in natural language is necessary and beneficial, but biology requires reasoning at both the microscopic and macroscopic level simultaneously. This work should explicitly address what differs between these domains rather than letting the reader assume RL's success in LLMs transfers to biology in the same form.
5. **Misleading abstract.** The abstract reads as a summary of the introduction rather than a full overview of the paper. It sets expectations the main text then has to correct, which puts the reader on guard throughout. A rewrite that covers the full scope would fix this.
6. **SFT and RL not compared on equal footing.** SFT trains on ~15k curated E. coli plasmids from PlasmidScope/Addgene for 3 epochs with gradient accumulation and warmup. RL uses no additional data but gets a reward signal. These are fundamentally different types of supervision. The authors should consider compute-matched comparisons, or discuss how much the SFT model might improve with more data, more epochs, or better curation.
7. **Diversity assessment lacks baselines.** RL diversity drops to 0.588 vs. 0.915 for the base model (Table 1). The paper's own limitations section (5.3) acknowledges RL reduces functional diversity (10 unique ORIs in base vs. 7 in RL). But the 0.588 vs. 0.915 comparison lacks external baselines -- the reader has no reference point for what "good" diversity looks like in this domain.
8. **GC content is not genuinely emergent.** The paper concedes GC content "is partially encoded by the reward function" (Section 4.2) because rewarded regions likely have typical GC content. This undermines the "emergent" framing for GC content specifically, yet the paper still presents it alongside genuinely emergent metrics (codon usage, Gibbs free energy) without distinguishing which properties are independent of the reward and which are partially correlated.
9. **Small evaluation sample.** The authors evaluate on only 50 rollouts per prompt across just 2 prompts (ATG codon and GFP cassette). This is the main factor limiting soundness.

---

> ### Author Rebuttal · Authors · 2026-03-30
>
> We thank the reviewer for the thorough and constructive evaluation. We address each weakness and question below, supported by substantial new experiments.
>
> W1 — No reward function ablation: Completed. We trained five ablation models (no repeat penalty, no length prior, no cassette bonus, length only, CDS only), each evaluated on 4,000 sequences. The MFE gradient across ablations (CDS-only −0.103 → length-only −0.126 → no-cassette −0.134 → no-repeat −0.141 → GRPO −0.149 vs. Addgene 500 reference −0.151) directly distinguishes emergent properties from indirect optimization. No single reward component produces real-plasmid-like thermodynamic stability. See Table A and our response to Reviewer MMB3 for the full ablation table and three-tier emergence taxonomy: [link](http://icml-rebuttal-tables-6f3393db.s3-website-us-east-1.amazonaws.com/)
>
> W2 — Optimized parameters lack causal validation: The ablation provides the requested causal evidence. Each model isolates a reward component, and the MFE gradient demonstrates that stability accumulates as structural constraints are added.
>
> W3 — Only GRPO tested: Early experiments with PPO showed slower convergence; we focused on GRPO accordingly. DPO was not considered as we targeted online methods, but remains future work.
>
> W4 — LLM-to-biology analogy too quick: Revised. We now explicitly address what transfers (structural reward signals can induce biophysical realism without direct optimization) and what does not (biology requires simultaneous reasoning across molecular and organismal scales; we evaluate only in silico properties, not biological function in a host).
>
> W5 — Misleading abstract: We will rewrite to include concrete numbers, the ablation finding, and the revised emergence taxonomy.
>
> W6 — SFT and RL not compared on equal footing: The rejection sampling baselines provide compute-matched comparisons. Rejection sampling (10,000 samples) from Base yields 2.8% pass rate; from GRPO, 64.6%. Best-of-16 selection from Base/SFT achieves 2.9%/2.8%. RL shifts the generative distribution and this cannot be replicated by sampling more from Base/SFT.
>
> W7 — Diversity lacks baselines: No established baselines exist for functional diversity in generative plasmid design — this is a new task. We provide two new reference points: (1) the Addgene 500 panel as an external anchor, and (2) rejection sampling from Base, which maintains 1.000 diversity but at 2.8% pass rate, confirming the valid plasmid subspace is narrow. The surprisal analysis (Section 4.4) provides the strongest evidence that diversity reduction reflects learning valid structure rather than pathological collapse.
>
> W8 — GC content not genuinely emergent: Agreed and reclassified as "partially correlated" in the revised taxonomy.
>
> W9 — Small evaluation sample: Now 8 prompts × 500 samples = 4,000 sequences per model, with per-prompt breakdowns.
>
> Q1: See ablation study above.
>
> Q2: In NLP, RL post-training typically degrades perplexity on held-out text — e.g., Ouyang et al. (InstructGPT) report increased perplexity after RLHF. Here, held-out log-probability improves slightly. This is unusual and may suggest the alignment tax is less pronounced when rewards are grounded in structural constraints rather than human preferences.
>
> Q3: Per-prompt analysis shows systematic variation. Pass rates range from 14% (pUC19 ORI prefix) to 97% (dual cassette), with most prompts achieving 55–88%. See Table B: [link](http://icml-rebuttal-tables-6f3393db.s3-website-us-east-1.amazonaws.com/)
>
> Q4: No pre-established baselines exist for this task. The base model's 0.915 diversity is misleading because 96.4% of those sequences fail QC, meaning that while they are diverse, it's mostly noise. We propose QC-filtered diversity as the appropriate metric. The Addgene 500 panel provides an external structural reference.
>
> Q5: Revised taxonomy: MFE = genuinely emergent (no single reward component produces it); codon usage = partially emergent (CDS reward contributes ~49%); GC content = partially correlated; ORF length = directly expected. See response to MMB3 for full ablation evidence.

---

> > ### Author Rebuttal · Reviewer_VevL · 2026-04-03
> >
> > Thank you for the detailed rebuttal and additional empirical evidence.
> > The additional ablation study addresses most of my prior concerns.
> >
> > Overall, this work would still benefit from a clearer presentation and unambiguous statement of key contributions, results, limitations, and future work. I’m maintaining my current rating (weak accept).

---

> > > ### Author Response · Authors · 2026-04-06
> > >
> > > We thank the reviewer for this feedback and fully agree that clearer presentation will strengthen the paper. We are restructuring the revision around three changes:
> > >
> > > 1. We replace "emergent biological realism" as the central framing with "effects of structural reward shaping on biophysical properties." The title and abstract will reflect this, stating concrete findings (MFE gradient, 71.6% pass rate, 22× sample efficiency over rejection sampling) rather than interpretive claims.
> > >
> > > 2. The introduction will state three numbered contributions: (i) first application of GRPO to whole-plasmid generation with an 8-prompt evaluation, (ii) a five-model reward ablation identifying the cassette bonus as the critical component and characterizing how biophysical properties improve as structural constraints compose, (iii) rejection sampling baselines establishing that RL provides genuine distributional shift unachievable by filtering.
> > >
> > > 3. We consolidate limitations into direct statements. The four-tier taxonomy (genuinely emergent / partially emergent / partially correlated / directly expected) replaces all unqualified uses of "emergent" throughout.
> > >
> > > We believe these changes address the reviewer's remaining concern and thank them for pushing us toward a more precise presentation.

---

### Decision · Program_Chairs · 2026-04-30

**Decision:**

Accept (regular)

**Comment:**

The reviewers agreed that the paper is generally sound and, after rebuttals, supported by reasonable experimental evidence to suggest practical value. There are remaining concerns about methodological novelty, which in my reading are valid but not sufficient to preclude acceptance.

One reviewer remains concerned about the loss of diversity in generations compared to the base DNA language model. In my reading, the most useful metric here is the number of unique passing generations from each model, and it appears that the proposed RL method does indeed greatly increase this. For a more rigorous evaluation of whether the loss of diversity could be improved, I would recommend the authors to baseline against the diversity of natural or designed plasmids.